# Application of Chemical Crystallization Circulating Pellet Fluidized Beds for Softening and Saving Circulating Water in Thermal Power Plants

**DOI:** 10.3390/ijerph16224576

**Published:** 2019-11-19

**Authors:** Ruizhu Hu, Tinglin Huang, Tianwei Wang, Huixin Wang, Xiao Long

**Affiliations:** 1Key Laboratory of Northwest Water Resource, Environment and Ecology, MOE, Xi’an University of Architecture and Technology, Xi’an 710055, China; www_lonely_com@163.com; 2Shaanxi Key Laboratory of Environmental Engineering, Xi’an University of Architecture and Technology, Xi’an 710055, China; 3Department of Production and Technology, Hebei Guohua Dingzhou Power Generation Co., Ltd., Dingzhou 070334, China; luckyone345@126.com (T.W.); SDtangzhch@163.com (H.W.); 4Department of Physical and Chemical Detection and Analysis Department, HEBEI Ji-Yan Energy Science and Technology Research Institute Co., Ltd., Shijiazhuang 050001, China; paul8122@126.com

**Keywords:** circulating pellet fluidized bed, thermal power plant, circulating cooling water, concentration ratio, zero liquid discharge

## Abstract

The circulating pellet fluidized bed (CPFB) softening method is a highly efficient and environmentally friendly softening technology that can be used to reduce water hardness during the pretreatment process of circulating water in thermal power plants. The performance of chemical crystallization CPFB reactors was tested for increasing the concentration ratio and softening the circulating water in a thermal power plant in Dingzhou, Hebei. The results show that usage of CPFB reactors removed water hardness and Ca^2+^ ions with efficiencies exceeding 60% and 90%, respectively. The size of the particles discharged from the reactors was approximately 1–3 mm, and the content of CaO in these particles was found to be greater than 50%. All the discharged particles were reused in the desulfurization system in the power plant. The operational cost of the CPFB system is US$0.074 per cubic meter of water. After adopting the proposed CPFB softening method in the Dingzhou Power Plant, the concentration ratio of the circulation cooling water was increased from 4.5 to more than 9. In addition, the amount of replenished water and sewage discharge were both reduced by 150 m^3^/h, and the amount of scale inhibitor used in the system was reduced by more than 30%. These improvements contribute to approximately US$200,000 in annual savings in the power plant. In summary, the CPFB softening method demonstrated a high hardness removal rate, strong economic benefits, and remarkable environmental and social benefits. Therefore, this method seems ideal for softening replenished circulating cooling water, increasing the concentration ratio of the water and achieving zero liquid discharge (ZLD) in thermal power plants.

## 1. Introduction

Thermal power generation is the primary mode of power generation in China. The installed capacity and power generation of thermal power plants in China account for approximately 80% and 78.72% of the total installed capacity and power generation in the country, respectively [1].

Thermal power plants consume large amounts of fresh water during the power generation process [2]. It has been reported that the water consumption of thermal power plants accounts for 11% of the total industrial water consumption in the country, and the circulating cooling water system of thermal power plants itself accounts for 84% of the water consumption of thermal power plants [3,4]. Therefore, reducing the consumption of circulating cooling water is of great significance for realizing zero liquid discharge in power plants in accordance with the national zero liquid discharge policy. The amount of circulating cooling water used can be reduced by changing the water source and reducing freshwater consumption [1]. For example, Yi et al. proposed using reclaimed water as circulating cooling water to reduce freshwater consumption and improve the utilization efficiency of water resources [5]. Reductions in the amount of circulating cooling water used can also be realized by improving the concentration ratio and utilization rate of the circulating water. For example, in a study carried out by Rahmani et al., the authors demonstrated that increasing the concentration ratio of circulating water from 6.5 to 9 can reduce annual freshwater consumption by 1.1 × 10^6^ m^3^. Such an approach can greatly reduce the amount of water used for replenishing circulating water as well as that of wastewater discharge [6].

Fouling is the first problem that needs to be solved to improve the concentration ratio of circulating water. Currently, fouling issues are primarily resolved by either adding chemical agents or using sulfuric acid as a scale inhibitor. Ochoa et al. experimentally studied the performance of a specific type of corrosion and scale inhibitor by adding these chemical inhibitors to a circulating water system [7]; it is common to use scaling inhibitors like phosphates, polyphosphates, and organophosphonates, as well as corrosion inhibitors such as zinc sulphate and azoles [8]. Rahmani et al. analyzed the scale-inhibiting effect and the associated working principle of adding sulfuric acid as a scale inhibitor in the fluid during the pretreatment process for circulating water replenishment [8]. Sulfuric acid has normally been used for carbonate calcium scale control in cooling water systems, as acid treatment converts calcium bicarbonate to more stable and soluble calcium sulfate. Although these two methods can effectively prevent the formation of calcium carbonate scales, failing to remove calcium carbonate from the circulating water results in very high hardness in the wastewater generated in subsequent processes. Therefore, it becomes more challenging to fully treat wastewater in later stages. Adding lime or sodium carbonate directly is an effective way to soften water, which is a different method from those mentioned above and can reduce the hardness of water. However, this method will produce a lot of sludge, and the operation is complex and equipment fouling occurs easily. Thus, this pretreatment method is not suitable for softening of circulating water [9,10].

The chemical crystallization circulating pellet fluidized bed (CPFB) softening method is a highly efficient and environmentally friendly softening technology that can be used to reduce water hardness during the pretreatment process of the circulating water. For example, Hu et al. applied CPFBs to soften groundwater and determined the operating parameters under specific water quality conditions [11]. They further studied the growth rate of the particles in Amsterdam’s crystallization fluidized bed reactor, as well as the relationship between the growth rate of the bed, rising flow rate, particle size, and supersaturation conditions on a pilot scale. A mathematical model for calculating the corresponding parameters is also provided in their study [12]. Tang et al. applied chemical-crystallization CPFBs to treat wastewater in a thermal power plant and verified their water-softening effect [13]. 

In summary, CPFBs can be used in the softening pretreatment of circulating water to remove calcium carbonate and prevent scale formation, as well as to simplify subsequent wastewater treatment processes. As an engineering case of circulating-water saving technology applied in a large-scale power plant, this paper provides important engineering data and information for the subsequent application of the proposed technology in industrial water circulation systems. In this study, the softening performance of CPFBs was studied under production testing conditions on the circulating water used in the Guohua Dingzhou Power Generation Co., Ltd., Hebei, China. Additionally, the impact of water quality was explored on improving the concentration ratio of circulating water after softening pretreatment under laboratory conditions. 

## 2. Materials and Methods

### 2.1. Material

In this study, the production tests were performed on the replenishment of circulating water at Guohua Dingzhou Power Generation Co., Ltd. Water was sampled directly from the reservoir for our tests. Table 1 summarizes the water quality parameters. A NaOH solution with a concentration of 30% was dispersed in the test water. The pH value of the test water was tuned using a H_2_SO_4_ solution with a concentration of 92%. Crystal seed size ranged from 0.4 to 0.6 mm, and their density was 3.93 g/cm^3^.

### 2.2. Full-Scale Experimental System and Process Description

Figure 1 shows a flow chart of the circulating-water replenishment and softening system under consideration. This system is designed to handle a flow rate of 1200 m^3^/h using three circulating pellet fluidized bed (CPFBs) reactors with a diameter of 2400 mm each. Each reactor is designed to perform water treatment at a flow rate of 400 m^3^/h. The system also includes a set of lye- and acid-dosing systems, one set of seed-dosing systems, and one set of particle-collecting systems. The total area of the workshop is 170 m^2^.

To replenish the circulating water, fresh water is first injected into the CPFB system and then discharged into the circulating-water pool, accompanied by the disposal of calcium carbonate particles. Details of the operating procedure and reaction processes can be found in existing literature [11]. The calcium carbonate particles disposed by the system are transported directly from the collection tank to the power plant’s desulfurization system to be used as a desulfurizer.

The water discharged from the CPFB system after pH adjustment was used to perform dynamic scale-inhibition simulation tests and scale-inhibition reagent-reduction tests in the laboratory. The test apparatus is shown in Figure 2. The test methods and evaluation standards used were determined based on Chinese chemical industry standards [14,15]. During the entire test process, the ΔB is calculated and kept to less than 0.2:(1)$$\Delta B=K_{Cl^{-}}-K_{Ca^{2+}}\leq 0.2$$
where $K_{Cl^{-}}$ is the concentration of chloride ion, and $K_{Ca^{2+}}$ is the concentration ratio of calcium hardness.

Finally, a corrosion test was performed using the water sample at a high concentration ratio. The corresponding test methods and evaluation standards were determined based on Chinese national standards [16]. The determination of corrosion rate (B) is expressed as the annual corrosion rate (mm/a), and the calculation method is as follows. According to the standards, the corrosion rate of all kinds of stainless-steel equipment used in circulating cooling water system should be of less than 0.005 mm/a:(2)$$B=3650\frac{G}{AT\rho }$$
where G is the weight loss of test piece (g), T is the test time (d), A is the area of test piece (cm^2^), and ρ is the density of the metal (g/cm^3^).

### 2.3. Analytical Methods

The total hardness and concentration of Ca^2+^ and Mg^2+^ were determined via the EDTA titration method [17]. Particle size was determined via the ASTM screening method [18]. The pH value of the water was measured using both an online and real-time pH meter (HACH sc200) as well as a handheld portable pH meter (HACH HQ11d). The composition of the particles emitted from the system was characterized using an inductively coupled plasma optical emission spectrometer (Optima 8000, Perkin Elmer), an electronic balance (ME104/02, Mettler Toledo), and a microwave digestion instrument (Multiwave PRO, Anton-Paar). A scale-inhibition tester [14] and a ZJ-type corrosion rate tester [16] were used to measure scale inhibition. 

## 3. Results and Discussion

### 3.1. Softening Performance of the CPFBs for Replenishment of Circulating Water

#### 3.1.1. Efficiency of CPFBs for Reducing Hardness

Figure 3 shows the softening effects of the three CPFBs on the discharge water under different time and flow rate conditions. All three fluidized beds were operating at a load of 50–110 m/h. As shown in Figure 3, the total hardness of the discharged water could reach as much as 1.4 mM, and the fluidized beds could reduce total hardness by 40–50%. In addition, the concentration of Ca^2+^ ions was approximately 0.4 mM, and the fluidized beds could remove up to 90% of the Ca^2+^ ions. It can be seen from the figure that the total hardness and Ca^2+^ ions concentration in the water increased slightly after adjusting the pH with acid. This is because a small volume of fine calcium carbonate particles flows out with the water during the crystallization process and these therefore fail to crystallize on the surface of the seed crystal. However, the addition of sulfuric acid causes the calcium carbonate to dissolve, which results in a slight increase in total hardness and Ca^2+^ ion concentration [10].

Because the HCO_3_^−^ ion content in the water was higher than the Ca^2+^ ion content, total hardness and the removal efficiency of Ca^2+^ ions are directly related to the amount of NaOH added to the water. A greater NaOH dosage results in higher removal rates, and therefore higher operating costs. As shown in Figure 4, the total cost of the chemicals (NaOH + H_2_SO_4_) per ton of water ranges between $0.06 and $0.10 when the removal rate of Ca^2+^ varies between 75% and 90%. During actual operation, the hardness removal effect must be adjusted to reduce costs [11].

#### 3.1.2. Efficiency of CPFBs for Reducing Alkalinity

Figure 5 shows the alkalinity removal performance of the three CPFBs on the discharged water under different time and flow rate conditions. All three fluidized beds were operated at a load of 50–110 m/h. A conversion process between HCO_3_^−^ ions and CO_3_^2−^ ions in water can observed from the figure corresponding to before and after the pH was adjusted by the addition of acid. The HCO_3_^−^ ion content of the discharged water ranged between 2 and 3 mM, and 30%–40% of the HCO_3_^−^ ions were removed by the fluidized beds. The amount of HCO_3_^−^ ions removed from water is proportional to the amount of Ca^2+^ ions removed and is in accordance with the theoretical chemical formulation.

As shown in Figure 5, HCO_3_^−^ ions first reacts with the NaOH in the water to form CO_3_^2−^ ions and water. The majority of the CO_3_^2−^ ions then react with the Ca^2+^ ions in the water to form calcium carbonate (CaCO_3_), which crystallizes on the surface of the seed crystal. A small volume of calcium carbonate flows out with the water before it can begin to crystallize, and a part of the CO_3_^2−^ ions will remain in the water, making it have a high pH. This is because the concentration of CO_3_^2−^ ions reaches the solubility product constant of calcium carbonate during its formation. Because the amount of Ca^2+^ ions in the water is limited, the CO_3_^2−^ ions are not consumed completely. After adding H_2_SO_4_ to the water, the CO_3_^2−^ ions are converted into HCO^3−^ ions again, causing the pH of the water to drop to between 7 and 8. Owing to the high residual alkalinity of the water, there still exists a risk of scaling at high concentration ratios. Therefore, it is necessary to further add a small amount of H_2_SO_4_ to prevent scaling in the subsequent circulating water system.

#### 3.1.3. Composition Analysis of Particles Discharged from the CPFBs

The only by-product of the circulating fluidized bed softening system is crystallized calcium carbonate particles, which can be used in a variety of applications [19]. One example of application is as a desulfurizer in the desulfurization systems in power plants. During desulfurization, the desulfurization agent, CaCO_3_, decomposes into CaO, which reacts with sulfur dioxide to form calcium sulfate [20]. Therefore, there are strict requirements on the CaO content in the desulfurizer particles. 

Figure 6 shows a scanning electron microscopy (SEM) image of the particles discharged from the system after about 20 days of equipment operation, and their size is usually approximately 1.0–3.0 mm at the time of discharge [21]. Via testing and analysis, the content of CaO in the particles generated from the CPFBs in the Dingzhou Power Plant was found to be 51.10%, as shown in Table 2. This content satisfies the minimal CaO content requirement of 50% in the particles intended for use as desulfurizer in the Dingzhou Power Plant [22] and can therefore be re-used in the desulfurization systems in the plant. Since mid-September 2018, the particles produced by the CPFBs have been re-used in the desulfurization system in the Dingzhou Power Plant, thus achieving zero emissions in the entire softening system.

### 3.2. Application of a CPFB System for Water Saving and Emission Reduction in Power Plants

#### 3.2.1. Dynamic Simulation Test on the Circulating Water

Increasing the concentration ratio of circulating water has been identified as an effective means for saving water and reducing emission in the circulating water systems of power plants [1,6]. To further investigate this approach, the dynamic scale-inhibition test was performed first using the water discharged from the CPFB system as the target sample. Based on the test results, a corrosion test was conducted using rotating coupons in order to optimize the production control parameters. The results of the dynamic scale-inhibition test are shown in Table 3.

As shown in Table 3, scale and corrosion inhibitors were added to the water discharged from the CPFB system in the Dingzhou Power Plant to obtain a concentration of 8.5 mg/L. In addition, the phenolphthalein alkalinity (p-alkalinity) of the water was maintained below 1.0 mM by adding sulfuric acid to the circulating water. These two approaches can prevent the formation of scales at a concentration ratio (K_limit_) as high as 10.82. Even after multiplying by a safety factor of 0.85, a safety concentration ratio of 9.20 is attained. At such a high concentration ratio, the corrosion test was conducted and the corrosion rate of TP316L stainless steel was 0.0005 mm/a as calculated by Equation (2), which satisfies the corrosion rate requirements of stainless-steel equipment [15,23]. Therefore, these tests can be used to determine effective circulating water control parameters for power plant operation, as shown in Table 4.

#### 3.2.2. Effect of Water Discharged from Circulating Pellet Fluidized Bed on Dosage of Scale Inhibitors

A scale-inhibitor reduction test was performed on water discharged from the CPFB system in the Dingzhou Power Plant. The test results are shown in Table 5. After adding a corrosion and scale inhibitor to the circulating water up to a concentration of 8.5 mg/L, the corrosion rate of TP316L stainless steel was determined to be 0.0003 mm/a as calculated by Equation (2) at a safety concentration ratio (85% K_limit_) of 5.64. After reducing the concentration of the corrosion and scale inhibitor to 6.0 mg/L and controlling the p-alkalinity of the water to be less than 1.0 mM by adding sulfuric acid, the corrosion rate of TP316L stainless steel was determined to be 0.0008 mm/a as calculated by Equation (2) at a safety concentration ratio (85% K_limit_) of 9.27. Under such conditions, no significant corrosion would occur on stainless-steel equipment [15].

According to these test results and upon combining them with the actual operation conditions on site, it is suggested to adopt the second set of operating conditions listed in Table 5. Under these conditions, the concentration ratio of the circulating water can be increased to 9.20, and the dosage of the scale inhibitor can be reduced by 30%.

#### 3.2.3. Impact of High Concentration Ratio on Promoting Zero Emissions in the Entire Plant

The ultimate goal of increasing the concentration ratio of the circulating water system in the power plant is to save water and achieve zero wastewater discharge. The amount of fresh water replenished to and the amount of wastewater discharged from the circulating water when using different concentration ratios can be obtained by calculating the volume of water in the circulating water system under different conditions, as shown in Figure 7. It can be seen from the figure that under the current working conditions (with a concentration ratio of 4.5 for the circulating water) in the Dingzhou Power Plant, the amount of water replenished to and the amount of wastewater discharged from the circulating water system are 1200 m^3^/h and 199 m^3^/h, respectively. When the concentration ratio was increased to 9.2, the amount of water replenished to and the amount of wastewater discharged from the circulating water system became 1050 m^3^/h and 49 m^3^/h, respectively. Both the amount of replenishing and discharging water were reduced by 150 m^3^/h. In addition, the total volume of wastewater discharged from the circulating system was reduced to 49 m^3^/h, which can be fully treated by the desulfurization system. Therefore, no wastewater would be discharged into the environment.

### 3.3. Evaluation of Economic and Environmental Benefits of CPFBs System

Increasing the concentration ratio of the circulating water can greatly reduce freshwater consumption and alleviate water stress in the local area. Reducing the waste content discharged from the circulating water system can be an effective approach to achieving zero liquid discharges and therefore reducing environmental pollution. In addition, the reduction of the required dosage of scale inhibitor in the circulating water can simplify the subsequent processes required to handle materials that are difficult to treat using scale inhibitors (such as phosphorus). Meanwhile, a reduction in the use of scale inhibitors also helps to protect the environment [24]. Through a comprehensive calculation, we found that Dingzhou Guohua Power Generation Co., Ltd. can save up to approximately 200,000 US dollars (US$) per year after increasing the concentration ratio of the circulating water using the crystallization softening system (as shown in Table 6). Therefore, the proposed softening technology can provide significant economic and environmental benefits.

## 4. Conclusions

(1)The application of circulating pellet fluidized beds to soften the circulating water in the Dingzhou Power Plant results in a hardness removal rate of 40%–50% and a Ca^2+^ ion removal of 90%, both of which ensure stable quality of the softened water. In addition, CaO contents in the discharged particles are greater than 50%, which allows these particles to be used directly in the desulfurization system in the power plant. Therefore, no wastewater or waste solids are generated from the entire system.(2)By pretreating the water via the use of circulating pellet fluidized beds and by discharging it into the circulating water system in the Dingzhou Power Plant, the concentration ratio of the circulating water is increased from 4.5 to 9.2, the amount of replenishing water and wastewater discharges are both reduced by 150 m^3^/h, and the dosage of the scale inhibitor is reduced by more than 30%.(3)The application of the circulating crystallization pellet fluidized bed system to soften the circulating water in the Dingzhou Power Plant reduces the cost of treating the circulating water to only 0.072 US$/m^3^. The power plant can thus save as much as 200,000 dollars per year. Therefore, the proposed softening technology can provide significant economic, environmental, and social benefits.

## Figures and Tables

**Figure 1 ijerph-16-04576-f001:**
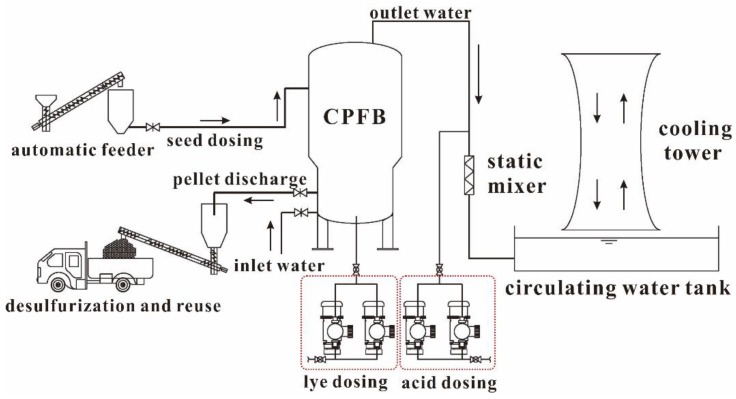
Flow chart of the circulating pellet fluidized bed softening system.

**Figure 2 ijerph-16-04576-f002:**
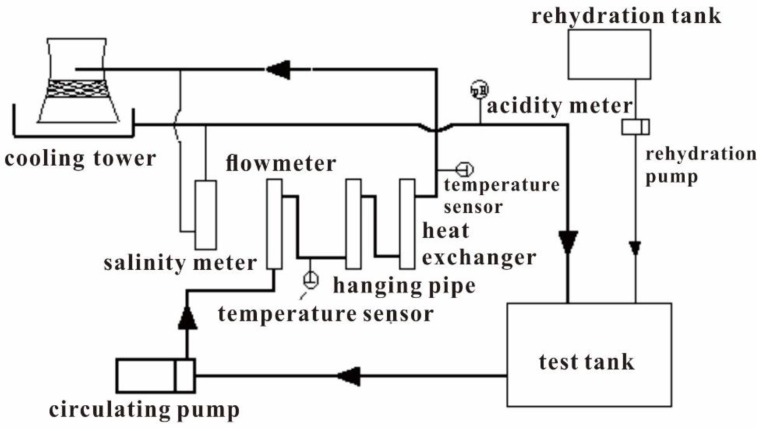
Flow chart of the dynamic scale-inhibition simulation test.

**Figure 3 ijerph-16-04576-f003:**
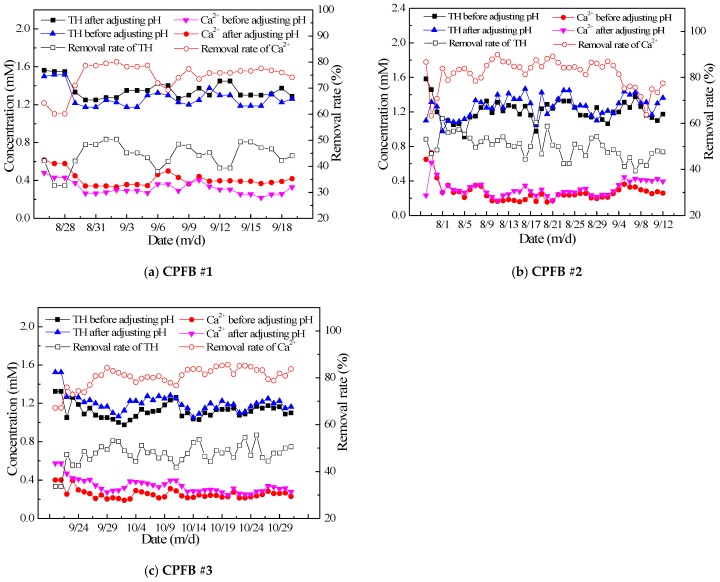
Hardness and Ca^2+^ ion removal performance of circulating pellet fluidized beds (CPFBs). ((**a**)-hardness and Ca^2+^ ion removal performance of circulating pellet fluidized bed(CPFB) #1 at different dates; (**b**)-hardness and Ca^2+^ ion removal performance of circulating pellet fluidized bed(CPFB) #2 at different dates; (**c**)-hardness and Ca^2+^ ion removal performance of circulating pellet fluidized bed(CPFB) #3 at different dates).

**Figure 4 ijerph-16-04576-f004:**
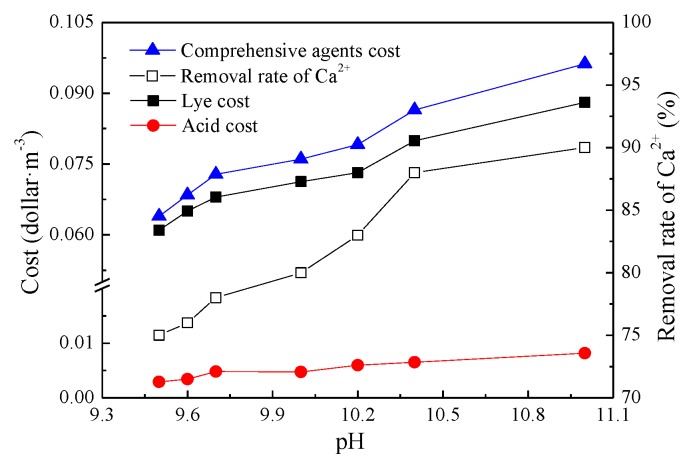
Dosage and cost of the chemicals used in the system under different pH conditions.

**Figure 5 ijerph-16-04576-f005:**
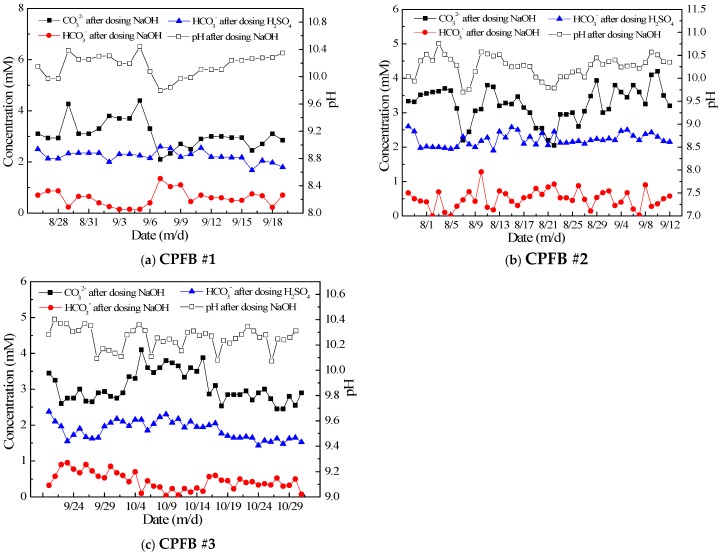
Alkalinity removal performance of the CPFBs. ((**a**)-alkalinity removal performance of the circulating pellet fluidized bed(CPFB) 1# at different dates; (**b**)-alkalinity removal performance of the circulating pellet fluidized bed(CPFB) 2# at different dates; (**c**)-alkalinity removal performance of the circulating pellet fluidized bed(CPFB) 3# at different dates).

**Figure 6 ijerph-16-04576-f006:**
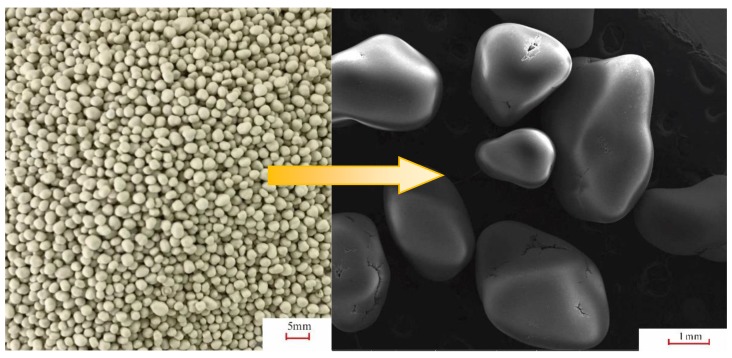
SEM image of the particles discharged from the system.

**Figure 7 ijerph-16-04576-f007:**
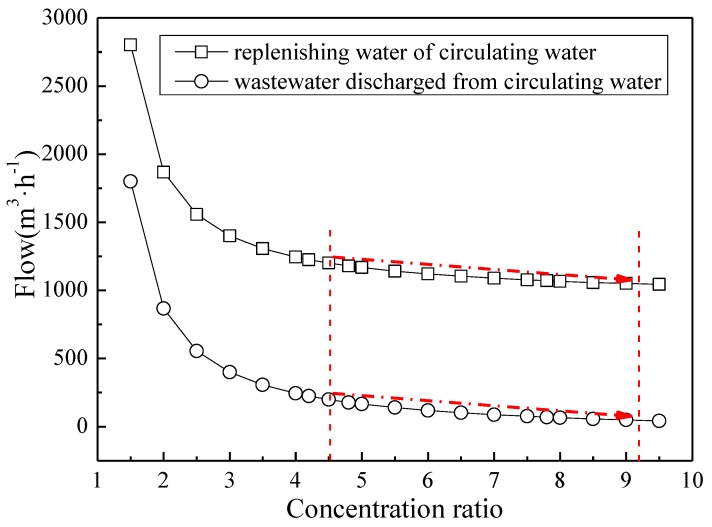
Amount of water replenished to and amount of wastewater discharged from the circulating water system for different concentration ratios.

**Table 1 ijerph-16-04576-t001:** Summary of the water quality parameters of the circulating water.

pH	Turbidity (NTU)	Total Alkalinity (mM)	HCO_3_^−^ (mM)	Total Hardness (mM)	Ca^2+^ (mM)	Mg^2+^ (mM)
7.5–7.7	<1	3.1–3.5	3.1–3.5	2.3–2.75	1.6–1.75	0.7–1.0

**Table 2 ijerph-16-04576-t002:** Composition analysis results of the particles generated by the CPFB.

Items	MgO	Al_2_O_3_	SiO_2_	K_2_O	MnO_2_	Fe_2_O_3_	CaO
	(Determined Based on the Oxide Product of Different Elements)						
Analysis results	0.74%	0.07%	2.99%	0.09%	0.06%	1.33%	51.10%

**Table 3 ijerph-16-04576-t003:** The results of dynamic scale-inhibition test.

Water Sample	K_limit_	85% K_limit_	Cl^−^_limit_ (mg/L)	Ca^2+^_limit_ (mg/L)	ΔB	Note
Water discharged from the CPFBR system	10.82	9.20	238	92.58	−0.20	Chemicals were added to the water sample up to a concentration of 8.5 mg/L; acid was added to the water sample to control the p-alkalinity (≤1.0 mM)

**Table 4 ijerph-16-04576-t004:** Suggested control parameters for circulating water.

Indicator	Concentration Ratio	Calcium Ion (mg/L)	Hardness (mM)	P-Alkalinity (mM)	Total Alkalinity (mM)	pH
Circulating water	≤9.20	≤77.28	≤20.08	≤1.0	≤6.6	≤8.75

**Table 5 ijerph-16-04576-t005:** The results of scale-inhibition reduction test.

Operating Conditions	K_limit_	85% K_limit_	Cl^−^_limit_ (mg/L)	Ca^2+^_limit_ (mg/L)	ΔB	Control Conditions
1	6.64	5.64	146	52.99	0.19	Scale inhibitor concentration = 8.5 mg/L
2	10.91	9.27	240	91.18	−0.18	Scale inhibitor concentration = 6.0 mg/L, p-alkalinity ≤ 1.0 mM, controlled by adding acid

**Table 6 ijerph-16-04576-t006:** Evaluation of the economic benefits (US dollars).

Number	Indicator	Quantity	Unit Cost	Economic Benefit (dollars per year)	Note
1	Reduction in replenishing water (m^3^/h)	150	$0.398 per m^3^	480977	Operating for 335 days a year
2	Reduction in waste discharge (m^3^/h)	150	$0.072 per m^3^	87134	Operating for 335 days a year
3	Reduction in scale inhibitor dosage (mg/L)	2.5	$1.156 per ton	22499	Average water usage rate = 1000 m^3^/h
4	Amount of water replenished to the circulating water (m^3^/h)	1050	$0.072 per m^3^	390150	Current real water volume 6,722,400 m^3^/y
5	Cost savings per year			200459

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
