# Peer review of "Application of Chemical Crystallization Circulating Pellet Fluidized Beds for Softening and Saving Circulating Water in Thermal Power Plants"

_ijerph, 2019, doi:10.3390/ijerph16224576_

Round 1

Reviewer 1 Report

This paper aims at softening and saving circulating water in thermal power plants. It is interesting but some revisions are needed.

(1) The logic should be enhanced. For example, “Circulating pellet fluidized beds (CPFBs) are a highly efficient and environmentally friendly softening technology that can be used to …”.  BEDS ARE A …? CPFBs ARE TECHNOLOGY? Please note that CPFBs are devices but not technology.

(2) The authors stated “being the first case study of circulating-water saving technology applied in a large-scale power plant in China”. I doubt it is the first one. Even if you are true, what do you intend to say if it is the first case study only in China?

(3) How were the data in Table 1 obtained?

(4) Line 119: “According to the standards, stainless steel equipment should be less than 0.005 mm/a.” Please check. What of the stainless steel equipment?

Author Response

Reply to the reviewer 1

1) The logic should be enhanced. For example, “Circulating pellet fluidized beds (CPFBs) are a highly efficient and environmentally friendly softening technology that can be used to …”.  BEDS ARE A …? CPFBs ARE TECHNOLOGY? Please note that CPFBs are devices but not technology.

Reply:

Thank you very much for your comments which help us improve the quality of our manuscript. There appear to be some discrepancies in the logic of the article, and I have made changes to resolve these. (page 1, line 14,24,29,30-31; page 2, line 74).

(2) The authors stated “being the first case study of circulating-water saving technology applied in a large-scale power plant in China”. I doubt it is the first one. Even if you are true, what do you intend to say if it is the first case study only in China?

 Reply:

We investigated and visited hundreds of power plants under the jurisdiction of five major domestic power groups and local power companies but found no application of this technology for saving circulating water, so we have reason to believe that this is the first case of large-scale use in a large-scale domestic power plant.

Saving of circulating water has also been implemented in recent years in China, mainly in response to the zero discharge policy of waste water proposed by the government. Here, it is proposed as the first case mainly to provide reference for treatment effect, operation cost, operation management, etc., for subsequent engineering cases.

(3) How were the data in Table 1 obtained?

Reply:

The data in Table 1 is the raw water quality index, which is obtained by our actual test, and is the same as the water quality index provided by the power plant.

 (4) Line 119: “According to the standards, stainless steel equipment should be less than 0.005 mm/a.” Please check. What of the stainless steel equipment?

Reply:

The stainless steel mentioned here refers to all kinds of stainless steel used in the circulating cooling water system. We understand that the statement is unclear and have thus revised it in the text (page 4, line 129-130).

Reviewer 2 Report

This manuscript explains about the pretreatment method used in softening of the circulating water in cooling tower of power plants. Higher removal total hardness and calcium ion removal efficiencies could achieved by using the CPFB technology. it also manuscript explains about the reducing water discharge and scaling issues was discussed. 

Introduction:

Authors should improve the introduction by using the generality of application 

The authors need to include more details about the other pretreatment methods used with the hardness removal rate and removal efficiencies. Though two references (Ochoa et al and Rahmani et al) were mentioned but these does not complete the overall methods used. line 52, include the time need to save the amount of water. 

Also, along with CPFB technology, authors need to discuss about the advantages and disadvantages other technologies used.

Results and discussion:

Figure titles should be explained with details. Also, the authors need to add few sentences about the trends of decrease or increase of ions concentrations. please include how was the removal rate ( Both calcium ions and total hardness) was calculated? In Figure 5, for CFPB#3 the removal rate of HCO3- was increased instead of decreasing at start, cite the reason for that. 

Figure 6 shows the discharged particles size, however it didn't mention about the time used for the discharge since the growth of particles depends on the time of discharge. Remove the scale bar with the other details from the obtained sem image, manually just add the scale bar. its good to add a figure showing the time vs increase in the size of the particles discharged to understand better about the increase in removal rate of calcium ions. 

The manuscript needs proof reading with the native English speaker. The manuscript could be accepted after including the revisions mentioned. 

Author Response

Reply to the reviewer 2

The authors need to include more details about the other pretreatment methods used with the hardness removal rate and removal efficiencies. Though two references (Ochoa et al and Rahmani et al) were mentioned but these does not complete the overall methods used. line 52, include the time need to save the amount of water. 

Reply:

Thank you very much for your comments. The details of pretreatment technology have been supplemented in this paper (page 2, line 60-61, 64-65, 69-73) referring to the following two articles. Line 52: water saving amount refers to a year, which is marked in red in the paper (page 2, line 53).

1.Vahedi A, Gorczyca B. Application of fractal dimensions to study the structure of flocs formed in lime softening process[J]. Water Research, 2011, 45(2):0-556.

2.O'Donnell A J, Lytle D A, Harmon S, et al. Removal of strontium from drinking water by conventional treatment and lime softening in bench-scale studies[J]. Water Research, 2016, 103:319-333.

Also, along with CPFB technology, authors need to discuss about the advantages and disadvantages other technologies used.

Reply:

The advantages and disadvantages of adding the scale inhibitor and sulfuric acid are marked in red in this paper (page 2, line 66-68). Another pretreatment technology and its advantages and disadvantages are added as well (page 2, line 69-73).

Results and discussion:

Figure titles should be explained with details. Also, the authors need to add few sentences about the trends of decrease or increase of ions concentrations. please include how was the removal rate ( Both calcium ions and total hardness) was calculated? In Figure 5, for CFPB#3 the removal rate of HCO3- was increased instead of decreasing at start, cite the reason for that. 

Reply:

Table 1 shows the concentration of calcium ion and total hardness in the inlet water of the fluidized bed. According to figure 3, we can see the concentration of calcium ion and hardness in the outlet water at different times and pH values. Through the comparison of the concentration of calcium ion in the inlet water and outlet water, we can calculate the removal rate of calcium ions and total hardness.

As Figure 5 shows the concentration of calcium ions at different times and pH values, the calcium ions in the outlet water every day are related to the operation, and the concentration of HCO3- decreases with the increase in NaOH dosage, which is not directly related to the time. The curve only shows the daily effluent concentration.

Figure 6 shows the discharged particles size, however it didn't mention about the time used for the discharge since the growth of particles depends on the time of discharge. Remove the scale bar with the other details from the obtained sem image, manually just add the scale bar. its good to add a figure showing the time vs increase in the size of the particles discharged to understand better about the increase in removal rate of calcium ions. 

Reply:

There is no mention of the time considered for the discharge and the SEM pictures of particle size at different times, because previous research papers have already studied this (reference 10). The target of study in this paper is production equipment. The CPFBs operation needs to follow the regulation of the power plant, which is not convenient if one wishes to study the change of particle size with time in detail, and it only provides the time at which the final particles are discharged. I have supplemented it appropriately (page 7, line 203) and I will try my best to overcome the difficulties in the future research and the study will be more detailed.

The manuscript needs proof reading with the native English speaker. The manuscript could be accepted after including the revisions mentioned. 

Reply:

The paper has been polished by professional institutions.

Reviewer 3 Report

This paper investigates CFPB technology which could soften replenished circulating cooling water in thermal power plants. A case study is presented in terms of experimental and economic analysis. It is generally well written and organised. Here are some minor comments before it could be accepted.

1.      The writing of this paper lacks flow and the coherence of space using is not good. Please double check.

2.      In introduction, the last two paragraphs are suggested to be merged into one.

3.      What I mainly concern is that the National Key Research and Development Program usually adopts the mature or good technology which has been well investigated. In this case, what is the innovative part of this paper since the CFPB maybe quite common in the power plant application. I really suggest that the author should conduct a comparative analysis of the current and previous work on CFPB technology. Thus, the novelty of this paper could be further illustrated. Or a simplified simulation is also welcome.

Author Response

Reply to the reviewer 3

1.The writing of this paper lacks flow and the coherence of space using is not good. Please double check.

Reply:

Thank you very much for your comments. I have modified flow and the coherence of space of the article with the help of a professional institute.

2.In introduction, the last two paragraphs are suggested to be merged into one.

Reply:

I have merged the last two paragraphs into one.

3.What I mainly concern is that the National Key Research and Development Program usually adopts the mature or good technology which has been well investigated. In this case, what is the innovative part of this paper since the CFPB maybe quite common in the power plant application. I really suggest that the author should conduct a comparative analysis of the current and previous work on CFPB technology. Thus, the novelty of this paper could be further illustrated. Or a simplified simulation is also welcome.

 Reply:

Fluidized beds are widely used in industrial production, but the circulating pellet fluidized bed is our patented invention under the support of the National Key Research and Development Program.

We investigated and visited hundreds of power plants under the jurisdiction of five major domestic power groups and local power companies but found no application of this technology for saving circulating water, so we have reason to believe that this is the first case of large-scale use in a large-scale domestic power plant. In this case, the innovative aspect of this study is that it provides a reference for treatment effect, operation cost, operation management, etc., for subsequent engineering cases.

The simulation work of the CPFBs is in progress.

Round 2

Reviewer 1 Report

(1) In the Response, the authors reclaimed that their study “Being the first case study of circulating-water saving technology applied in a large-scale power plant in China”. In regard of this, firstly, you may say the chemical crystallization circulating pellet fluidized bed you are investigating on is the first one, but not circulating-water saving technology. There are plenty of literature focused on the circulating-water saving case study. Secondly, please be noted that this journal is an international journal but not a Chinese one. International journals publish papers with novelty in all the world, rather than novel only in a specific country. If the authors plan to publish their paper in an international country, I strongly suggest revise this sentence.

(2) Line 129: The authors changed “stainless steel equipment should be less than 0.005 mm/a” to “all kinds of stainless steel equipment used in circulating cooling water system should be of less than 0.005 mm/a.” It is still unclear “what” of all kinds of stainless steel equipment? Do you intend to mean “the corrosion rate of stainless steel” if I guess correctly? Please ensure.

Author Response

(1) In the Response, the authors reclaimed that their study “Being the first case study of circulating-water saving technology applied in a large-scale power plant in China”. In regard of this, firstly, you may say the chemical crystallization circulating pellet fluidized bed you are investigating on is the first one, but not circulating-water saving technology. There are plenty of literature focused on the circulating-water saving case study. Secondly, please be noted that this journal is an international journal but not a Chinese one. International journals publish papers with novelty in all the world, rather than novel only in a specific country. If the authors plan to publish their paper in an international country, I strongly suggest revise this sentence.

Reply:

Thank you for your comments. I revised the sentence.(page 2, line 85-86).

(2) Line 129: The authors changed “stainless steel equipment should be less than 0.005 mm/a” to “all kinds of stainless steel equipment used in circulating cooling water system should be of less than 0.005 mm/a.” It is still unclear “what” of all kinds of stainless steel equipment? Do you intend to mean “the corrosion rate of stainless steel” if I guess correctly? Please ensure.

Reply:

Thank you for your comments. I revised the sentence. (page 4, line 129).

Reviewer 2 Report

Please add the scale bar manually for the SEM figure. there is no need to add other details on the figure. Article could be accepted after minor revisions.

Author Response

Reply:

Thank you for your comments. I add the scale bar manually for Figure 6. (page 7, line 212).